# Analysis of Using Machine Learning Techniques for Estimating Solar Panel Performance in Edge Sensor Devices

**Dalibor Dobrilovic, Jasmina Pekez \*, Visnja Ognjenovic and Eleonora Desnica**

Technical Faculty "Mihajlo Pupin" Zrenjanin, University of Novi Sad, 23000 Zrenjanin, Serbia; dalibor.dobrilovic@tfzr.rs (D.D.); visnja.ognjenovic@tfzr.rs (V.O.); eleonora.desnica@tfzr.rs (E.D.)
\* Correspondence: jasmina.pekez@tfzr.rs

**Featured Application: This paper presents a methodology for the implementation of edge intelligence for predicting solar panel performances on wireless sensor nodes.**

**Abstract:** The importance of the usage of renewable energy sources in powering wireless sensor nodes in IoT and sensor networks grows together with the increasing number of utilized sensor nodes. Considering the other types of renewable energy sources, solar power differs as the most suitable one and emerges as the major source for powering sensor nodes. Thus, the consideration of using sensor nodes and collected sensor data for estimating solar panel performances and therefore solar power potential can improve the efforts in this direction. This paper presents the methodology for implementing edge intelligence on wireless sensor nodes for solar panel output voltage estimation and forecasting. The methodology covers the usage of the Python Scikit-learn package and micromlgen library for the implementation of edge intelligence on Arduino clone-based sensor nodes, particularly the development boards based on the ESP8266 chips. Scikit-learn is used for analyzing the efficiency of various regressors on collected solar data. The micromlgen library is then used for implementing those regressors on Arduino and clone nodes. The prediction of solar panel voltage generation is based on a single-sensor reading—UV or BH1750 light sensor. The Random Forest and Decision Tree regressors are implemented on the ESP8266-based development board—Wemos D1 R2. The estimation accuracy of the RF model is an MSE of approximately 0.10, MAE of 0.07 for UV and 0.04 for BH1750, and an $R^2$ of approximately 0.93 for both UV and BH1750 light sensors. The Decision Tree model has a lower accuracy with an MSE between 0.13 and 0.14, MAE of 0.07 for UV and 0.04 for BH1750, and $R^2$ of 0.90 and 0.89 for the UV and BH1750 sensors, respectively. The methodology and its efficiency are presented and discussed in this paper.

**Keywords:** solar panel performance; solar power potential estimation; Arduino edge intelligence; Scikit-learn library; micrgenml library

## 1. Introduction

In recent years, we have witnessed the expansion of an edge computing approach in building complex and distributed IT systems. This expansion is accompanied by the growing tendency to implement machine learning (ML) in edge devices. Thus, edge computing (EC) with integrated ML facilitates the main goal of the EC paradigm, moving data processing to edge devices and reducing the latency caused by cloud-based processing. All these goals are further enhanced with machine learning techniques utilized to increase the efficiency of the edge nodes.

This paper deals with the importance of the usage of renewable energy sources in a particular field of interest covering IoT systems and sensor networks. This paper targets the problem of powering the sensor node, especially in complex systems. The importance of powering wireless sensor nodes in IoT and sensor networks grows together with the increasing number of utilized sensor nodes. The complexity of the system, the larger

number of nodes, and the dispersity and non-easy reachability of the locations where the sensor nodes are deployed increase the importance of efficient powering of sensor nodes. Considering the available types of renewable energy sources, solar power distinguished itself as the most suitable one and emerged as the major source for powering sensor nodes. So, the consideration of using sensor nodes and collecting sensor data for estimating solar panel performances and therefore solar panels and power potential can improve the efforts in this direction.

The utilization of artificial intelligence (AI) on sensor nodes results in the attainment of edge intelligence (EI). EI represents the data analysis and solution recommendations at the point where the data are generated or acquired, thus saving the time and sometimes bandwidth of the sensor networks. Generally, edge intelligence means the implementation of AI at the far end of IoT and sensor networks, more precisely on sensor nodes and microcontroller boards.

EI [1] can improve the process of the collection of solar radiation data and make the estimation of solar panel performance more efficient and more accurate. In addition, the utilization of edge intelligence can make solar radiation data collection more massive and can be enabled on the sensor nodes without solar panels.

In an effort to define and simplify the process of the efficient implementation of AI on sensor nodes in existing and future sensor networks, this paper presents the methodology for implementing edge intelligence on wireless sensor nodes for solar panel performance (voltage output) estimation and forecasting.

The AI on the edge devices is used to predict solar panel behavior depending on various ambient parameters (visible and UV light intensity, air temperature humidity, and solar panel temperature). AI-enhanced estimation should enable the estimation of solar panel performance at sensor stations without solar panels—with the usage of other sensors.

Generally, PV module performance parameters are evaluated based on I–V and P–V curves and numerous other parameters [2]. In this research, we used the open-circuit voltage ($V_{oc}$) parameter. $V_{oc}$ is measured under the standard test (STC) or real-time operating conditions. It is measured with a voltmeter or voltage sensor when the panel is not connected to any equipment. The value of the voltage in this case is generally higher than the maximum voltage of the panel. There are several reasons for such an approach. First, open-circuit voltage ($V_{oc}$) is valuable for system planning to avoid overpowering electronics; in this case, it is interesting for the potential design of solar-powered sensor nodes. Second, it is interesting to investigate the influence of light intensity and temperature on solar panel performance and $V_{oc}$ is suitable for this research due to its dependency. The third reason is that we wanted to estimate solar panel behavior based on one output parameter, and we chose $V_{oc}$ because of all the enlisted reasons.

The methodology covers the usage of the Python Scikit-learn package and micromlgen library for the implementation of edge intelligence on Arduino sensor nodes. In the beginning, the process and platform designed for collecting solar radiation data are described in the paper. The Scikit-learn package is used for analyzing the efficiency of various regressors applied to collected solar data. The micromlgen library is then used for implementing those regressors on Arduino clone nodes, in this case on ESP8266-based sensor boards. The results of this implementation as well as its efficiency are presented and discussed in this paper.

The contribution of this paper presents the methodology for implementing edge intelligence for estimating the potential solar panel performance on the sensor nodes without solar panels, thus leading to the collection of valuable data for the potential redesign of non-solar-powered sensor nodes, at specific micro-locations, to solar-powered sensor nodes.

The difference in the methodology presented in this paper compared to other solutions is that all other solutions use field real-time measurements, with actual sensors for physical parameters that are monitored. Our solution differs because it uses regression methods implemented in edge devices (sensor nodes) to predict the values of physical parameters

without existing sensors for that specific parameter. In our proposal, the edge devices have implemented various regression methods to predict parameters of non-existing sensors, based on existing sensors installed for other purposes.

This paper is organized as follows. After the introduction section, the state of the art is presented. In the next section, the proposed methodology is presented in detail. Then, the results of the proposed implementation method are presented and discussed. Finally, the conclusion and further work are discussed.

## 2. Related Works

According to [3], the growth, development, and popularity of the IoT and big cloud services caused the need for edge computing. Edge intelligence or edge AI represents the combination of edge computing (EC) and distributed computing. In such environments, data processing is relocated from the cloud to the network edge. The goal is to implement computing and data storage nodes on edge devices, e.g., mobile devices or sensors. This goal is facilitated by the rapid changes and development in recent years [4]. As stated in [5], the main significance of EC is as follows:

- The combination of sensor technology and edge devices impacts efficient real-time data acquisition, especially in smart buildings and smart home systems;
- There is evident growth in edge-oriented communication technologies, e.g., device-to-device (D2D) communication;
- EC is targeted as an important component of edge intelligence, helping in reducing the response time, network traffic, and latency and saving energy and bandwidth;
- Edge devices store temporary real-time information;
- There is the possibility of extensive use of mobile phones as the edge devices;
- Edge devices can achieve significant collaboration taking advantage of their mobility, proximity, and deployment.

There are numerous examples of using edge intelligence for monitoring and managing solar panels. In [6], EI enables the integration of solar energy into the electrical grid and solves problems related to solar energy production. A remote monitoring system with an integrated artificial neural network (ANN) for detecting shading in photovoltaic panels is presented in [7]. Paper [8] presents a system that uses Siamese-twin neural networks for anomaly detection. The system is implemented in a solar farm on edge devices based on a Raspberry PI, Nvidia Nano, and Google Coral. Authors from [9] propose a framework for the detection of anomalies in decentralized photovoltaic plants. This proposal contributes to hybridization between edge, fog, and cloud layers.

Because Arduino-based clones are used in this research, it is important to show how popular Arduino is in academic institutions for research and education. The authors of [10] justified the popularity of Arduino development boards by analyzing the number of Arduino-based scientific papers. The research covers the period from 2010 to 2020 and shows constant growth in the platform's usage.

Further, the examples of the implementation of AI on Arduino-based development boards are presented in the following papers. Paper [11] proposes a real-time non-intrusive load classification (RT-NILC) IoT-based system with an Arduino-based data acquisition system. In [12], a small-scale two-wheel system connected to a control unit is developed using an ARDUINO Uno Rev3 microcontroller and a Support Vector Regression (SVR) model.

The Arduino UNO boards have certain limitations considering their memory and processing capabilities. In addition, they do not have integrated network connectivity modules. The low-cost Arduino-clone development boards based on ESP8266 chips, such as NodeMCU and equivalent ESP32-chip-based boards, have integrated Wi-Fi, and in the case of ESP32, Wi-Fi and Bluetooth Low Energy (BLE) connectivity. Paper [13] proposes a low-cost parking management system. The system is based on the Alibaba Cloud platform, machine learning, and an ESP8266-based Arduino-clone development board as a control module. In addition, [14] presents a similar ESP866 Wi-Fi module garbage management system for monitoring flame and several other ambient parameters. In [15], Node MCU

ESP8266 is used to monitor the quality of soil and predict crop types suitable for the monitored location [15]. ESP8266 is also used in the system for recognizing the character gestures [16]. Here, KNN (K-Nearest Neighbours), Decision Tree, and SVM (Support Vector Machine) classifications are implemented. The same development board is used in the system for real-time weather prediction systems [17].

Similar but newer and more advanced ESP32-based development boards implemented with ML are used in the following projects. The authors present the Wi-CaL crowd counting and localization system [18] with implemented machine learning (ML) and deep learning (DL). ESP32 also used machine learning in [19] for distance estimation appliances in real time. Paper [20] analyses the feasibility of deploying deep networks on ESP32 devices with TensorFlow. Research presented in [21] proposes the use of IoT and ML in hydroponic systems. The goal of the system is to enhance the growth of Holy Basil. The research [22] presents the Respiration DT (ResDT) model based on Wi-Fi Carrier State Information (CSI). The model uses machine learning for the monitoring and classification of patient respiration.

Furthermore, it is interesting to investigate the utilization of ML in evaluating and estimating solar panel performance. The authors of [23] suggest a decision-making model data processing technique and machine learning. The research findings justify the implementation of data science and machine learning in a solar PV panel cleaning system. Paper [24] introduces semi-supervised learning and one-class classification methods based on autoencoders. The presented methodology improves the nonlinear data representation of solar behavior. The authors in [25] evaluated more than 100 research articles to investigate ML implementation in solar cell fabrication. The findings show that the Random Forest (RF), linear regression (LR), XGBoost, and artificial neural network (ANN) algorithms are the most commonly employed techniques. Further, research results show XGBoost's superior performance in optoelectronic prediction, while RF, LR, and ANN algorithms are better suited for predicting electrical parameters. The review of existing machine learning (ML) approaches used in PV power forecasting, focusing on short-term horizons, is presented in [26]. The overview contains factors affecting solar PV power forecasting.

It is interesting to examine the tools for implementing ML in edge devices. TinyML supports running machine learning at embedded edge devices with limited processor and memory resources. The important issue for edge devices is power consumption, which should be minimal. So, TinyML enables migration to low-power IoT-based embedded edge devices and allows the development of novel applications without the need for processing on the cloud [27–29]. The overview of TinyML benefits is given in [30]. Authors in [30] conclude that TinyML is considered a promising AI alternative for extremely low-profile devices. The article [31] presents prediction methods based on artificial neural network (ANN) models. TinyML allows importing pre-trained ML models on the edge devices, thus achieving ML-as-a-Service (MLaaS). Paper [32] presents a TinyMLaaS (TMLaaS) architecture that presents several design variations in terms of energy consumption, security, privacy, and latency.

The implementation of TinyML is possible with the numerous frameworks. A good overview of TinyML frameworks is given in [28,29], and benefits are given in [33]. One of these frameworks is Micromlgen [34], and this framework is used in this research. The contribution presented in this paper is a methodology developed for the efficient implementation of ML on sensor nodes to estimate solar panel performance. The estimation should be made on sensor nodes not having solar panels; thus, it will be applicable for already designed and deployed sensor nodes, which can be used for estimating the solar potential of the current sensor location and its feasibility for upgrading solar-powered sensor nodes. This method includes the definition of the toolset: Python, Scikit-learn package, micromlgen library, and ESP8266-based development boards. This paper evaluates the methodology by evaluating the model's accuracy. The proposed methodology is based on previous authors' experience with wireless sensor networks and industrial IoT presented in [35], the implementation of AI on Arduino clone boards [36], and solar radiation data acquisition [37].

### 3. Methodology

This paper presents the methodology for using Python and solar radiation data to implement edge intelligence on Arduino devices for estimating solar panel outputs. The methodology is presented in Figure 1.

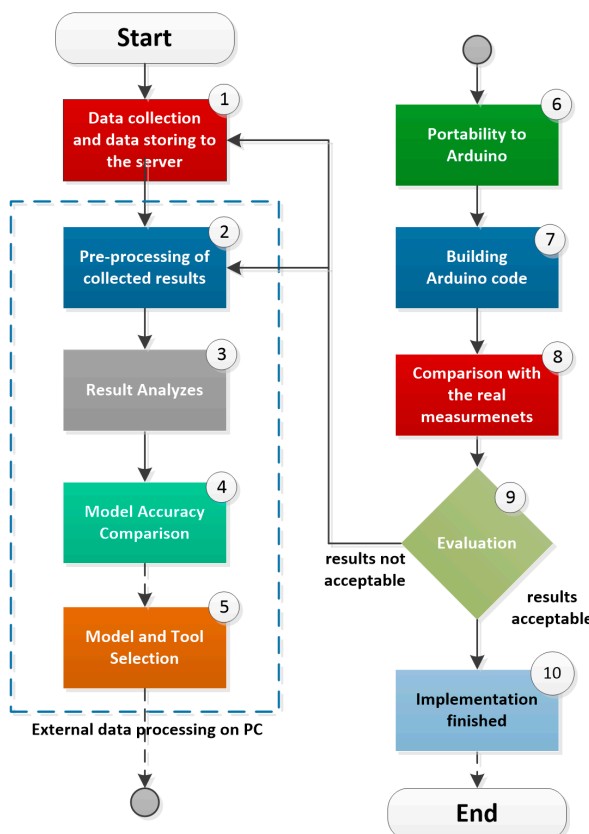

**Figure 1.** Methodology for implementing edge intelligence on Arduino and clone boards.

The proposed methodology for implementing AI on Arduino boards consists of the following steps:

(1) Initial solar radiation data collection with the platform that will be described in the following text.
(2) Pre-processing of collected results on PC (external) and preparation for further processing.
(3) Analyses of solar radiation data using Python and the Scikit-learn package.
(4) Model accuracy comparison, based on collected and processed results.
(5) The selection of the model to be implemented at the edge devices, in this case, Arduino clone platform based on ESP8266 chip, and selection of a tool for implementing the model.
(6) Building libraries for porting to the Arduino clone ESP8266-based platform using the selected tool.
(7) Building the code for the selected platform.
(8) Testing platform and comparison of the test results with real measurements.
(9) Evaluation of the results and comparison; if the results are not valid, return to step 1 or step 2 to correct the irregularities or to improve the process.
(10) If the results are valid, proceed to the implementation of the selected method.

To present the whole methodology, it is important to describe the following components: the solar data collection platform, Python and related packages used for implementing machine learning and different regressors (Numpy [38], Pandas [39,40], Scikit-learn [41], Micromlgen), and the process of evaluation using Arduino clone devices (in this case,

ESP8266-based microcontrollers). In addition, the Matplotlib [42] and Seaborn [43] libraries are used for data visualization.

### 3.1. The Platform for Solar Data Collection and Analyses

The platform for solar data collection is presented in another paper and described there in more detail [35], and it is shown in Figure 2.

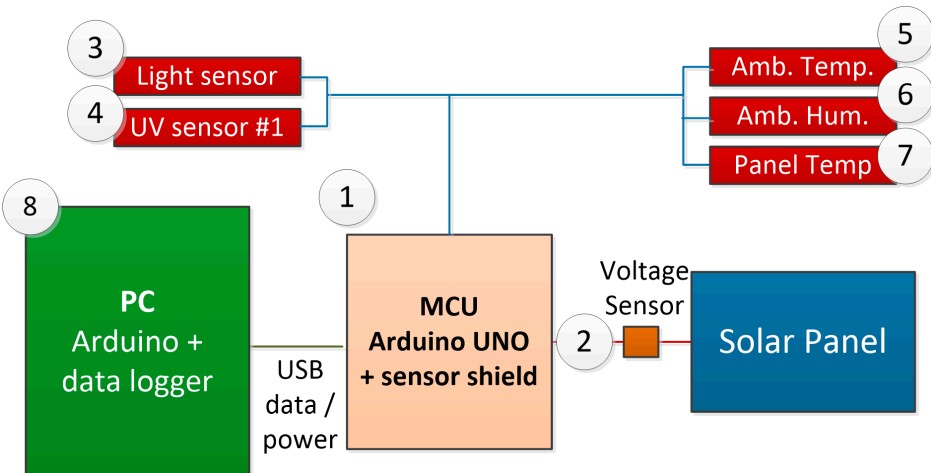

**Figure 2.** The solar data collection platform is based on Arduino.

The platform design based on Arduino for solar radiation data acquisition was as follows:

- Arduino UNO Rev3 (1);
- Voltage sensor (2);
- Light sensor—BH 1750 (3);
- UV sensor (4);
- DHT-22 temperature and humidity sensors (5) and (6);
- TMP36 temperature sensor for panel temperature (7);
- PC for external data processing (8);
- Solar panel: 81 × 137 mm 1.5 W 270 mA 5.5 V (9).

### 3.2. Collected Solar Data Analyses

The collected solar data were analyzed. The purpose of the analysis in this stage is to determine which sensor readings have the greatest impact on predicting solar panel output voltage. These analyses are based on using a Multi-Layer Perceptron regressor (MLP regressor) in Python with the Scikit-learn package. The Scikit-learn package supports more than 25 regressors, but the MLP regressor is chosen for the initial analyses following the positive experience and good results in previous analyses and works. Although generally unsuitable for non-linear regressions, $R^2$ is the standard method in Scikit-learn library metrics, and it can show which used methods can have non-linear results and can be particularly indicative when $R^2$ has negative values. Also, a certain number of models used in further research belong to the linear regression model family. We used the $R^2$ score in combination with other parameters such as MSE and MAE to detect suitable regressors for further implementation of AI. The results presented in Table 1 show that the best estimation can be achieved with the Lux sensor (BH1750) in combination with the UV sensor (RMSE of 0.08, MAE of 0.24, and R2 score of 0.9650) and with the Lux sensor only (0.08, 0.22, and 0.97, respectively). The next single sensor with the highest accuracy is the UV sensor (0.1, 0.29, and 0.94, respectively).

After the recognition of the estimation accuracy of various combinations of sensors separately, this research continues to explore the efficiency of the two most efficient sensors, BH1750 and UV. For this task, Python is used in combination with the Scikit-learn package.

**Table 1.** Comparison of various sensor combinations with MLP regressor in the estimation of solar panel output.

| Sc. No. | Input Data | $R^2$ | RMSE | MAE | EVS | Max. Err |
|---|---|---|---|---|---|---|
| 1 | Lux, UV_Raw, Temp_DHT22, Hum_DHT22, Temp_TMP36 | 0.9521 | 0.0904 | 0.2705 | 0.9529 | 0.2078 |
| 2 | Lux, UV_Raw, Temp_DHT22, Temp_TMP36 | 0.8926 | 0.1354 | 0.3156 | 0.8942 | 0.4926 |
| 3 | Lux, UV_Raw, Temp_TMP36 | 0.9535 | 0.0891 | 0.2662 | 0.9556 | 0.2113 |
| 4 | UV_Raw, Temp_TMP36 | 0.9581 | 0.0845 | 0.2583 | 0.9604 | 0.2122 |
| 5 | Lux, UV_ITeadRaw | 0.9618 | 0.0807 | 0.2401 | 0.9654 | 0.2460 |
| 6 | Lux | 0.9650 | 0.0772 | 0.2278 | 0.9661 | 0.3188 |
| 7 | Temp_DHT22 | 0.3548 | 0.3318 | 0.3843 | 0.3749 | 2.1943 |
| 8 | Hum_DHT22 | −0.3656 | 0.4827 | 0.4820 | 0.3109 | 2.6242 |
| 9 | Temp_TMP36 | 0.4444 | 0.3079 | 0.3475 | 0.4565 | 2.1791 |
| 10 | UV_Raw | 0.9424 | 0.0991 | 0.2895 | 0.9478 | 0.1995 |

## 4. Results

The results of using Python and 28 regressors supported by the Scikit-learn package are shown in Table 2. In the case of estimating the solar panel output voltage based on joint BH1750 and UV sensor readings and considering the $R^2$ score, Random Forest, MLP, and KNN regressors are identified as the most efficient regressors with a score of 0.95 and higher. The XGBoost, Gradient Boosting, Decision Tree, SVR, and Extra Trees are similarly efficient with a score of 0.90 or higher. The impact of the accuracy of the single-sensor-based estimation is slightly different. For example, for BH1750 only, the $R^2$ scores are 0.97894 for Random Forest, 0.96298 for K-Neighbors, 0.89068 for XGBoost, 0.61396 for SVR, 0.90218 for Gradient Boosting, 0.88959 for Decision Tree, and 0.95015 for Ex Trees. For UV single-sensor reading, the $R^2$ results for various regressors are 0.90521 for XGBoost, 0.95892 for Random Forest, 0.90604 for SVR, 0.90652 for Gradient Boosting, 0.95037 for MLP, 0.94961 for K-Neighbors, 0.89174 for AdaBoost, 0.90451 for Decision Tree, 0.90952 for Extra Trees, and 0.88147 for Voting. In both cases, the single-sensor impact is less accurate than the two-sensor estimation but still usable.

The $R^2$ score comparison for two sensors simultaneously is shown in Figure 3. The comparison of regressors is evaluated further with MSE and MAE metrics, as shown in the following figures.

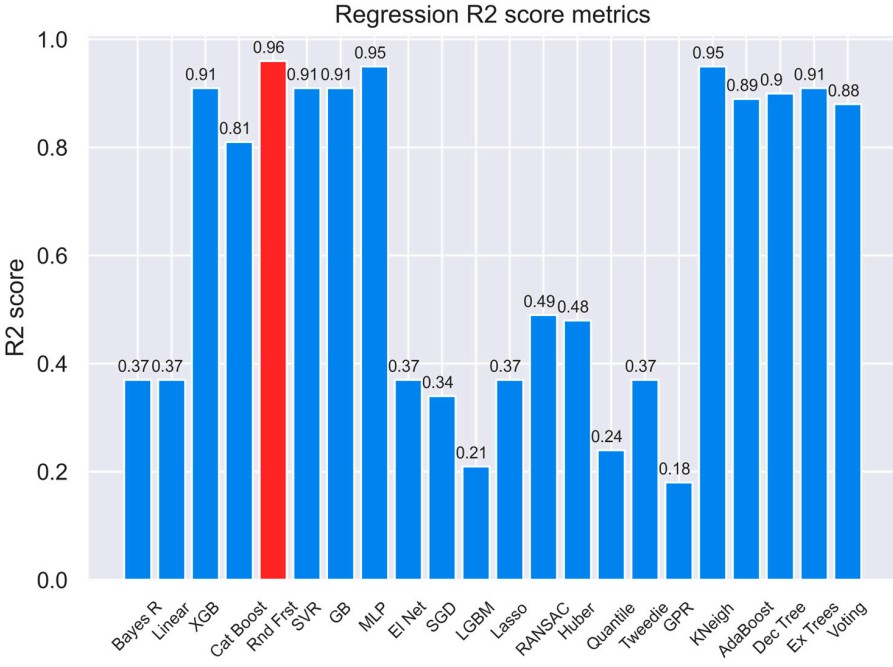

**Figure 3.** Comparison of $R^2$ score for different Scikit-learn regressors.

The comparison of MSE metrics is given in Figure 4.

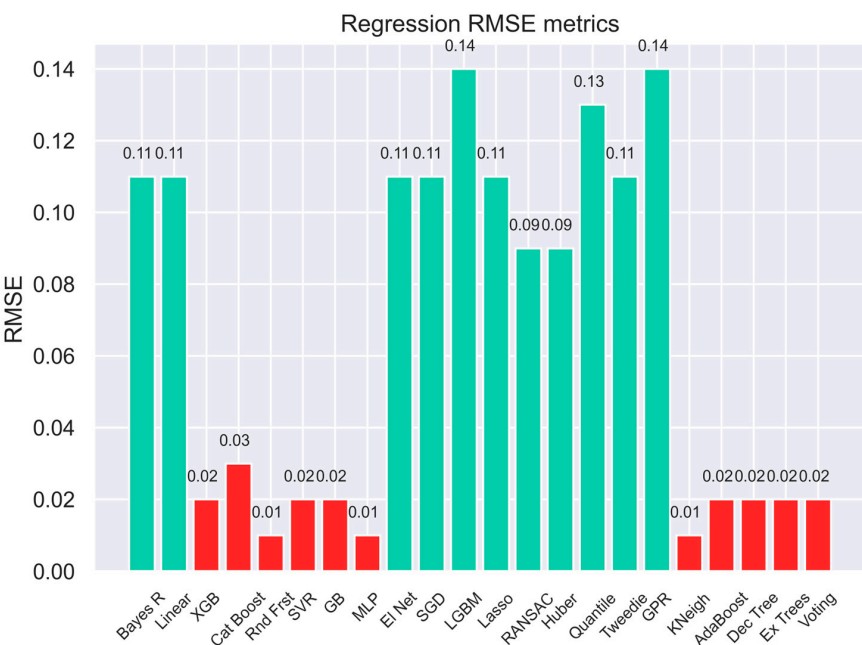

**Figure 4.** Comparison of MSE for different Scikit-learn regressors.

The comparison of MAE metrics is given in Figure 5.

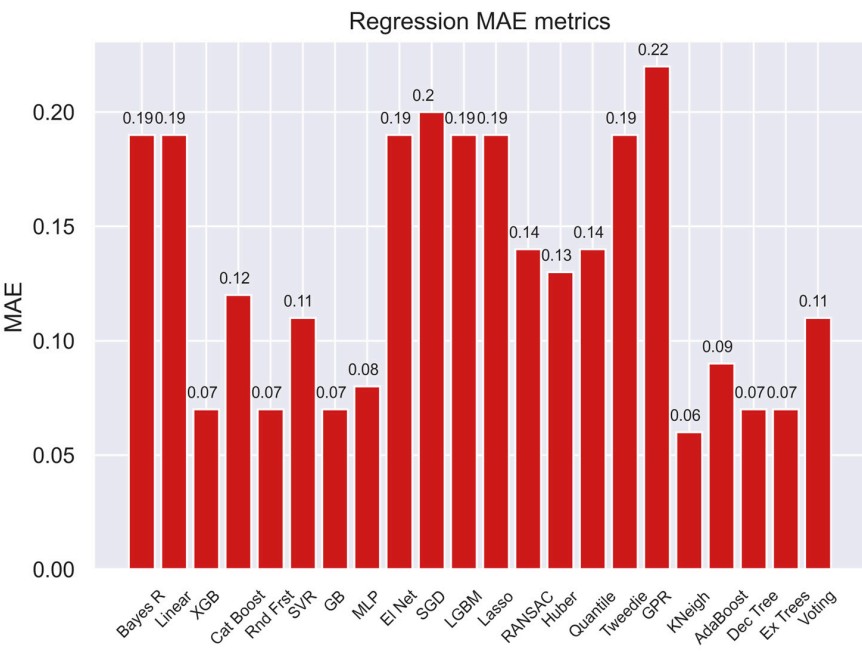

**Figure 5.** Comparison of MAE for different Scikit-learn regressors.

### 4.1. The Arduino Implementation

After the model accuracy comparison (phase 4), the next phase in the methodology is model and tool selection (phase 5). Considering the model accuracy and available tools for the implementation of tested models on the edge device, the Python library micromlgen is considered for this phase. The idea of this phase is to use the aforementioned machine-learning models as a tool for estimating solar panel output voltage. So, with this idea, we can use existing wireless sensor nodes equipped with visible light and UV sensors to estimate the solar potential of the location where the sensor node is deployed. The

intelligent sensor platform with the implemented AI and a reduced number of sensors and without solar panels is given in Figure 6.

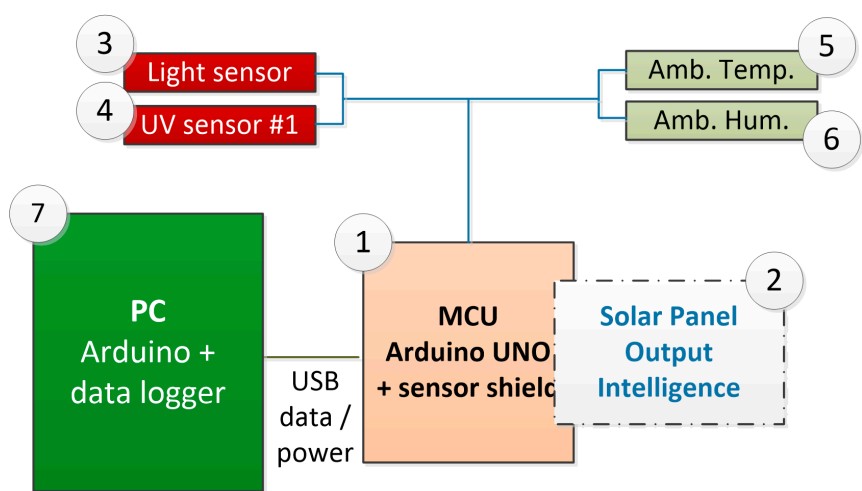

**Figure 6.** The sensor network platform with implemented AI.

**Table 2.** Comparison of various Scikit-learn supported regressors in the estimation of solar panel output based on UV sensor readings.

| Regression | R Score | $R^2$ Score | MSE | EVS | Max. Err | MAE | MSL | MAP | MedAE |
|---|---|---|---|---|---|---|---|---|---|
| Bayes R | 0.551882 | 0.371885 | 0.107154 | 0.413385 | 1.419959 | 0.190234 | 0.003582 | 0.038744 | 0.066796 |
| ARD | −0.00198 | −0.03552 | 0.176656 | $-2.22 \times 10^{16}$ | 2.594436 | 0.245601 | 0.00595 | 0.051628 | 0.205564 |
| Linear | 0.551602 | 0.370364 | 0.107414 | 0.411879 | 1.416933 | 0.190599 | 0.003588 | 0.038805 | 0.066233 |
| XGB | 0.96898 | 0.905209 | 0.016171 | 0.911202 | 0.705034 | 0.074994 | 0.000402 | 0.013361 | 0.050822 |
| Cat Boost | 0.858901 | 0.807068 | 0.032914 | 0.811086 | 0.669619 | 0.119967 | 0.001098 | 0.023656 | 0.100901 |
| Krnl Rdg | −7.52032 | −13.6857 | 2.505323 | −11.5554 | 4.595392 | 0.905294 | 0.192135 | 0.176634 | 0.232028 |
| Rnd Frst | 0.96632 | 0.958924 | 0.007007 | 0.961693 | 0.219079 | 0.066961 | 0.00016 | 0.011924 | 0.052773 |
| SVR | 0.942443 | 0.906048 | 0.016028 | 0.917416 | 0.277325 | 0.10573 | 0.000419 | 0.019669 | 0.109814 |
| GB | 0.96921 | 0.90652 | 0.015947 | 0.9123 | 0.699241 | 0.073867 | 0.000396 | 0.013157 | 0.050819 |
| MLP | 0.960045 | 0.950374 | 0.008466 | 0.955407 | 0.19942 | 0.07822 | 0.000195 | 0.01403 | 0.081658 |
| El Net | 0.552002 | 0.372548 | 0.107041 | 0.414041 | 1.421286 | 0.190074 | 0.00358 | 0.038717 | 0.067043 |
| SGD | 0.543944 | 0.339704 | 0.112644 | 0.413088 | 1.391616 | 0.198231 | 0.003718 | 0.040097 | 0.09443 |
| LGBM | 0.431142 | 0.206002 | 0.135453 | 0.250426 | 1.674311 | 0.187298 | 0.004571 | 0.038722 | 0.097639 |
| Lasso | 0.551684 | 0.370802 | 0.107339 | 0.412313 | 1.417802 | 0.190494 | 0.003586 | 0.038787 | 0.066395 |
| LARS | −0.00198 | −0.03552 | 0.176656 | $-2.22 \times 10^{16}$ | 2.594436 | 0.245601 | 0.00595 | 0.051628 | 0.205564 |
| RANSAC | 0.50913 | 0.492829 | 0.086522 | 0.49596 | 1.88031 | 0.136337 | 0.003298 | 0.02977 | 0.0827 |
| Theil–Sen | −5.46248 | −10.282 | 1.924665 | −8.75078 | 4.012532 | 0.805855 | 0.11491 | 0.156262 | 0.226623 |
| Huber | 0.46951 | 0.478527 | 0.088961 | 0.479205 | 1.997051 | 0.132971 | 0.003442 | 0.02956 | 0.090452 |
| Quantile | 0.183056 | 0.23943 | 0.129751 | 0.316501 | 2.459665 | 0.144412 | 0.004771 | 0.033602 | 0.058078 |
| Pass Agr | −16.7823 | −26.9486 | 4.767939 | −18.8068 | 3.865447 | 2.092106 | N/A | 0.380279 | 2.171411 |
| Tweedie | 0.551603 | 0.37037 | 0.107413 | 0.411885 | 1.416945 | 0.190597 | 0.003588 | 0.038805 | 0.066235 |
| GPR | 0.49625 | 0.181773 | 0.139587 | 0.260354 | 1.158412 | 0.222909 | 0.004401 | 0.044228 | 0.075856 |
| KNeigh | 0.96924 | 0.949614 | 0.008596 | 0.94964 | 0.355 | 0.063333 | 0.000202 | 0.011335 | 0.035 |
| Dummy | −0.00198 | −0.03552 | 0.176656 | $-2.22 \times 10^{16}$ | 2.594436 | 0.245601 | 0.00595 | 0.051628 | 0.205564 |
| Poisson | −0.0017 | −0.03552 | 0.176656 | $-2.22 \times 10^{16}$ | 2.594436 | 0.245601 | 0.00595 | 0.051628 | 0.205564 |
| AdaBoost | 0.960807 | 0.891735 | 0.01847 | 0.909818 | 0.71 | 0.093959 | 0.000453 | 0.016723 | 0.085143 |
| Dec Tree | 0.968853 | 0.904517 | 0.016289 | 0.910219 | 0.71 | 0.074304 | 0.000405 | 0.013235 | 0.050795 |

$R^2$ or coefficient of determination analyzes how differences in one variable can be explained by a difference in a second variable. Mean squared error (MSE) or mean squared deviation (MSD) calculates the amount of error in statistical models with the average squared difference between the observed and predicted values. When the MSE equals zero, the model has no error. The square root of the MSE calculates the root mean squared error (RMSE), giving the natural data units. MSE is analogous to the variance, and RMSE is analogous to the standard deviation. MAE (mean squared error) is the amount of error in your measurements and represents the difference between the measured value and "true" value, a risk metric corresponding to the expected value of the absolute error loss or -norm loss. EVS (explained variance score) or explained variation is used to measure the discrepancy between a model and actual data. Max. Err. (maximum error) calculates the maximum residual error, a metric that captures the worst-case error between the predicted value and the true value. MSL (mean squared log error) calculates a risk metric corresponding to the expected value of the squared logarithmic (quadratic) error or loss. MAP (mean absolute percentage error) or mean absolute percentage deviation (MAPD) is an evaluation metric for regression problems, sensitive to relative errors.

The micromlgen library is suitable for its utilization for several reasons. It is a well-documented library, with good examples. It is easy to implement, and it works with the Scikit-learn library, which is already used. The micromlgen library supports classifiers such as Decision Tree, Random Forest, XGBoost, Gaussian NB, Support Vector Machines (SVC and OneClassSVM), Relevant Vector Machines (from skbayes.rvm_ard_models package), and SEFR. The supported regressors are Decision Tree, Random Forest, Linear Regressor, and Logistic Regressor. The example Python code for building Arduino libraries for the first two regressors is shown in Listing 1 (phase 6). Decision Tree and Random Forest regressors are chosen because of their acceptably high accuracy, as shown in the results presented in Section 3, and because of their inclusion in the micromlgen library.

**Listing 1.** Section of Python script for building RF and DT regressor libraries.

```python
if __name__ == '__main__':
    regrRF = RandomForestRegressor(n_estimators=10, max_depth=10,
min_samples_leaf=5).fit(X_train, y_train)
    regrDT = DecisionTreeRegressor(ccp_alpha=0.0, criterion='squared_error',
max_depth=None,
            max_features=None, max_leaf_nodes=None,
            min_impurity_decrease=0.0,
            min_samples_leaf=1, min_samples_split=2,
            min_weight_fraction_leaf=0.0,
            random_state=0, splitter='best').fit(X_train, y_train)
with open('EdgeAI/RandomForestRegressor.h', 'w') as file:
    file.write(port(regrRF))
    with open('EdgeAI/DecisionTreeRegressor.h', 'w') as file:
    file.write(port(regrDT))
```

Code with the usage of the Micromlgen library is generated using Arduino IDE and compiled and uploaded to the ESP8266-based Wemos D1 R2 (phase 7) Arduino clone development board. Arduino UNO does not have enough capacity for storing and running this firmware, but ESP8266-based boards do. The ESP8266 board is useful for consideration due to its in-built Wi-Fi connectivity and usability for wireless sensor nodes in the network. In this experiment, a variant of the ESP8266-based sensor board, a Wemos D1 R2, is used.

### 4.2. ESP866 Results

The Micromlgen implemented library in Arduino accuracy compared to Python estimation accuracy is shown in Figures 7 and 8 for UV-based sensor estimation and in Figures 9 and 10 for BH1750-based sensor estimation. The figures show minor differences between Arduino and Python estimations.

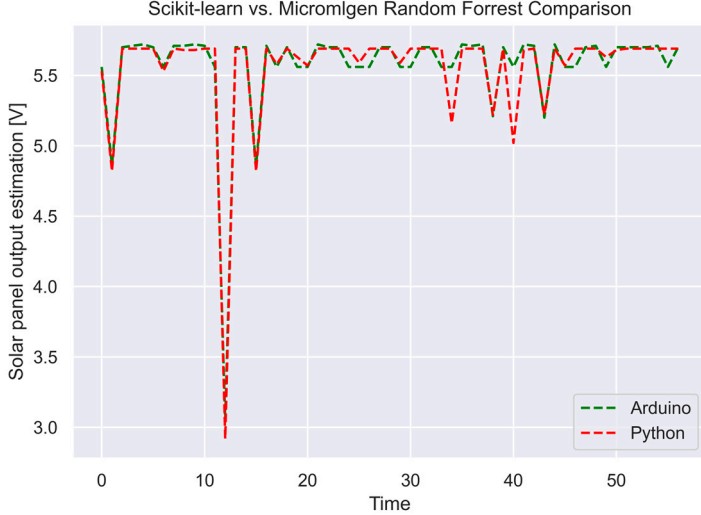

**Figure 7.** ESP8266 implemented RF regressor accuracy based on a UV sensor.

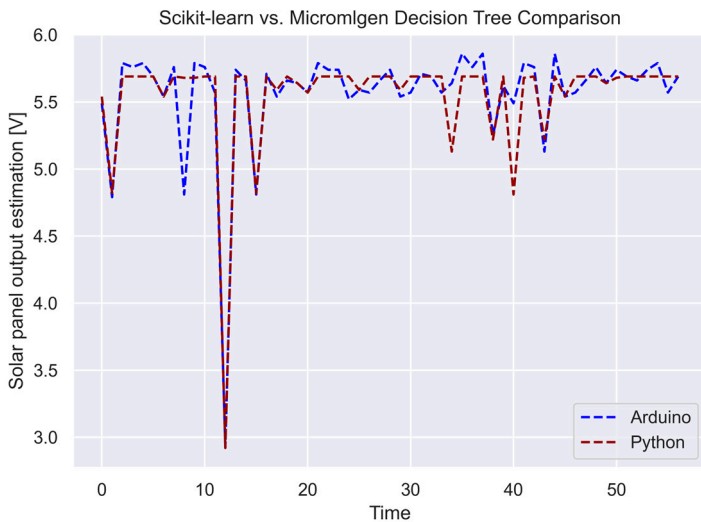

**Figure 8.** ESP8266 implemented DT regressor accuracy based on a UV sensor.

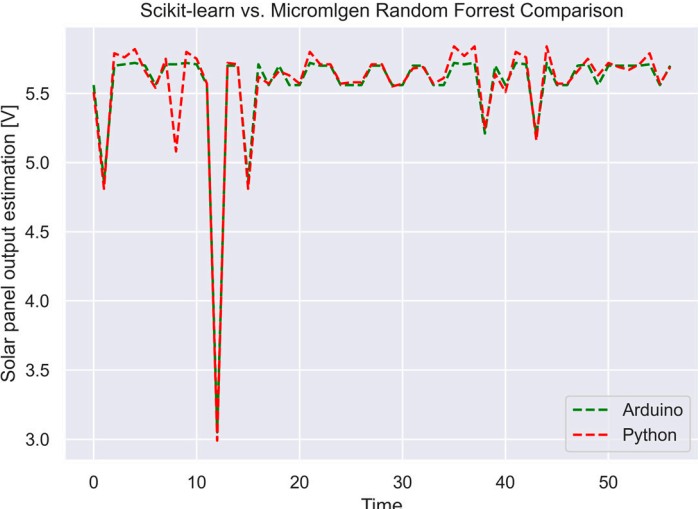

**Figure 9.** ESP8266 implemented RF regressor accuracy based on the light sensor.

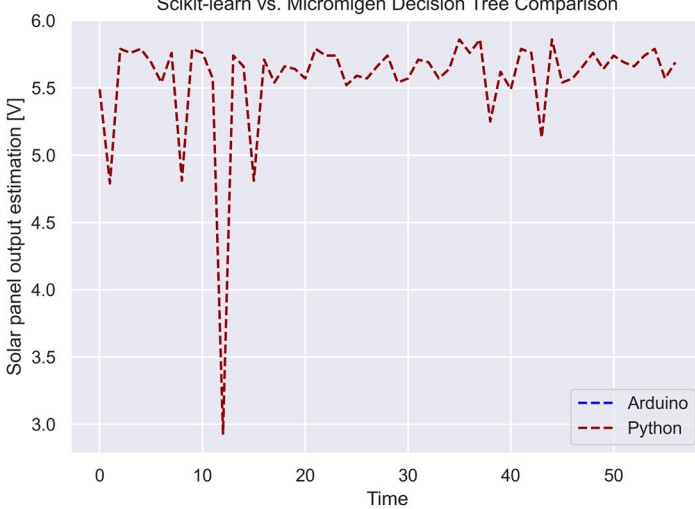

**Figure 10.** ESP8266 implemented DT regressor accuracy based on the light sensor.

Table 3 shows the comparison metrics of UV-based Random Forest and Decision Tree estimations, and BH1750-based (Lux) Random Forest and Decision Tree estimation, respectively. The current results show minor differences in measured values. The second and third columns give the ESP8266 estimation comparison to Python Scikit-learn estimated values (MSE and MAE), and the fourth and fifth give the ESP8266 comparison metrics compared to real measured values (MSE and MAE). The last column shows $R^2$ scores of ESP8266 estimation compared to measured values.

**Table 3.** Comparison of DT and RF accuracy implemented on the ESP8266 platform.

| Parameters | MSE Py | MAE Py | MSE Real | MAE Real | $R^2$ Real |
|---|---|---|---|---|---|
| UV RF | 0.10416 | 0.05263 | 0.10683 | 0.06947 | 0.93309 |
| UV DT | 0.17864 | 0.09456 | 0.127568 | 0.07404 | 0.90461 |
| Lux RF | 0.09789 | 0.05105 | 0.10278 | 0.03912 | 0.93808 |
| Lux DT | 0.00000 | 0.00000 | 0.13726 | 0.04509 | 0.88956 |

## 5. Discussion

After the implementation of Micromlgen regressors for ESP8266 devices, the comparison of the ESP8266 estimated and real values is analyzed and discussed. The $R^2$ values of both regressors (DT and RF) are considerably high; therefore, those results can be used for implementation in this stage of research. The ESP8266 Random Forest regressor has higher accuracy, with both sensors (an MSE of approximately 0.10, MAE of 0.07 for UV and 0.04 for Lux, and an $R^2$ of approximately 0.93). DT has slightly lower accuracy (MSE between 0.13 and 0.14, MAE of 0.07 for UV and 0.04 for Lux, and $R^2$ of 0.90 and 0.89 for UV and Lux sensors, respectively) in estimating real values compared to RF, but it is still accurate enough for justified implementation of both methods.

A limitation of the current research is the relatively balanced dataset, which was collected in a relatively short period, during sunny June days, resulting in a relatively small range of variances of the measured values. The collection of solar radiation data over much longer periods will be performed in the future and in further phases of this project.

The results of this research are important because they show that the proposed methodology is efficient enough to be implemented already at the sensor nodes. Thus, they can be deployed as an intelligent form of nodes, as shown in Figure 6, together with classical solar radiation data collectors presented in Figure 2.

## 6. Conclusions

The importance of the increase in alternative energy sources, especially solar power, reflects the field of powering electronic devices. In almost the last two decades, the number of electronic devices has increased multiple times, raising the problem of their electrical powering. This problem arises more with the introduction of wireless sensor networks, and even more with IoT and Smart technologies, such as Smart Cities, Smart Agriculture, Smart Manufacturing, etc. The problem we are facing can be solved with solar-powered sensor nodes. Therefore, it is very important to find a way for efficient powering of sensor nodes. This paper proposes a methodology that includes ML for the assessment of the solar panel performance and solar potential of the sensor node location, in cases when sensor nodes do not have solar panels.

In summary, this paper presents the methodology for implementing edge intelligence on sensor nodes. Edge intelligence helps in forecasting solar panel voltage generation. The methodology uses acquired solar data in building ML models to be implemented on microcontrollers. The set of tools includes Python, the Scikit-learn package, the micromlgen library, and ESP8266-based development boards. The proposed model predicts solar panel voltage generation based on a single-sensor reading using a UV or BH1750 light sensor. The Random Forest and Decision Tree regressors are implemented on the ESP8266-based development board—Wemos D1 R2. The estimation accuracy of the RF model is an MSE of approximately 0.10, an MAE of 0.07 for UV and 0.04 for BH1750, and an $R^2$ of

approximately 0.93 for both the UV and BH1750 light sensors. The Decision Tree model has a lower accuracy with an MSE between 0.13 and 0.14, MAE of 0.07 for UV and 0.04 for BH1750, and $R^2$ of 0.90 and 0.89 for the UV and BH1750 sensors, respectively. Both metrics justify the usage of the proposed methodology.

Further work should cover analyses of implementing ESP32 sensor boards, and their comparison with ESP8266 boards. Including multiple sensor readings for predicting voltage output values of solar panels. Finally, the comparison of other toolsets will be explored in further research.

**Author Contributions:** Conceptualization, D.D. and J.P.; methodology, D.D. and V.O.; validation, D.D., J.P. and E.D.; formal analysis, V.O. and E.D.; investigation, J.P. and E.D.; resources, D.D. and J.P.; data curation, D.D. and V.O.; writing—original draft preparation, D.D., J.P. and V.O.; writing—review and editing, E.D. and V.O.; visualization, D.D.; supervision, J.P. and E.D.; project administration, E.D.; funding acquisition, E.D. All authors have read and agreed to the published version of the manuscript.

**Funding:** This research was conducted through the project "Creating laboratory conditions for research, development, and education in the field of the use of solar resources in the Internet of Things", at the Technical Faculty "Mihajlo Pupin" Zrenjanin, financed by the Provincial Secretariat for Higher Education and Scientific Research, Republic of Serbia, Autonomous Province of Vojvodina, project number 142-451-3118/2023-01.

**Institutional Review Board Statement:** Not applicable.

**Informed Consent Statement:** Not applicable.

**Data Availability Statement:** The datasets presented in this article are not readily available because the data are part of an ongoing study.

**Conflicts of Interest:** The authors declare no conflicts of interest.

## Abbreviations

| | |
|---|---|
| ML | Machine Learning |
| DL | Deep Learning |
| AI | Artificial Intelligence |
| EI | Edge Intelligence |
| EC | Edge Computing |
| BLE | Bluetooth Low Energy |
| DT | Decision Tree |
| RF | Random Forest |
| SVM | Support Vector Machine |
| LR | Linear Regression |
| ANN | Artificial Neural Network |
| $R^2$ | Coefficient of Determination |
| RMSE | Root Mean Square Error |
| MAE | Mean Squared Error |
| EVS | Explained Variance Score |
| Max. Err | Maximum Error |
| MAE | Mean Absolute Error |
| MSL | Mean Squared Log Error |
| MAP | Mean Absolute Percentage Error |
| Notifications | |
| $V_{oc}$ | Open-circuit Voltage |
| UV | Analog value in the range 0–1023 of the UV sensor readings |
| Lux | BH1750 light sensor calculated value in lux units (lx) |
| Bayes R | Bayesian Ridge regressor |
| ARD | Automatic Relevance Determination (ARD) regressor |
| Linear | Linear regressor |
| XGB | Extreme Gradient Boosting (XGBoost) regressor |
| Cat Boost | CatBoost regressor |

| Krnl Rdg | Kernel Ridge regressor |
| Rnd First | Random Forest regressor |
| SVR | Support Vector Regression (SVR) regressor |
| GB | Gradient Boosting regressor |
| MLP | Multi-layer Perceptron (MLP) regressor |
| El Net | Elastic Net regressor |
| SGD | Stochastic Gradient Descent (SGD) regressor |
| LGBM | Light Gradient-Boosting Machine (LightGBM) |
| Lasso | Least Absolute Shrinkage and Selection Operator (Lasso) regressor |
| LARS | Least-angle regression (LARS) regressor |
| RANSAC | RANdom SAmple Consensus (RANSAC) regressor |
| Theil–Sen | Theil–Sen regressor |
| Huber | Huber regressor |
| Quantile | Quantile regressor |
| Pass Agr | Passive Aggressive regressor |
| Tweedie | Tweedie regressor |
| GPR | Gaussian Process (GPR) regressor |
| KNeigh | k-Nearest Neighbors regressor |
| Dummy | Dummy regressor |
| Poisson | Poisson regressor |
| AdaBoost | Ada Boost regressor |
| Dec Tree | Decision Tree |

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
