# Peer review of "Analysis of Using Machine Learning Techniques for Estimating Solar Panel Performance in Edge Sensor Devices"

_applsci, doi:10.3390/app14031296_

Round 1

Reviewer 1 Report

Comments and Suggestions for Authors

The authors of the paper presents the methodology for implementing edge intelligence on wireless sensor nodes for solar panel performance estimation and forecasting.

For the described methodology, all the phases should be inserted in one logical flowchart. In this way, the authors will increase the clarity of the paper.

The scientific contribution can be increased by inserting the adequate equations.

The contribution of the authors, by critical comparing the existing technologies, should be outlined.

Author Response

Dear Reviewer,

Thank you for your review of our manuscript. We have carefully considered your comments and believe that the quality of the paper has improved after incorporating your suggestions.

Below are our responses to your suggestions:

Comments and Suggestions for Authors

The authors of the paper presents the methodology for implementing edge intelligence on wireless sensor nodes for solar panel performance estimation and forecasting.

  • For the described methodology, all the phases should be inserted in one logical flowchart. In this way, the authors will increase the clarity of the paper.

We agree with the reviewer, and we are very thankful for this comment. We redesigned the Fig. 1 according to the comments.

1.2. The scientific contribution can be increased by inserting the adequate equations.

We agree with the reviewer for this comment. These are the following reasons why we omitted presenting formulas in the paper. The methods used in the paper are not based on formulas that can be simply presented. It is more difficult because of the number of methods we used in this paper.

1.3. The contribution of the authors, by critical comparing the existing technologies, should be outlined.

We agree with the reviewers comments, and therefore we inserted the following comment in the paper in the Introduction section.

“The difference in the methodology presented in this paper compared to other solutions is that all other solutions use field real-time measurements, with actual sensors for physical parameters that are monitored. Our solution differs because it uses regression methods implemented in edge devices (sensor nodes), to predict the values of physical parameters without existing sensors for that specific parameter. In our proposal, the edge devices have implemented various regression methods to predict parameters of non-existing sensors, based on existing sensors installed for other purposes.”

Submission Date

29 December 2023

Date of this review

09 Jan 2024 14:32:05

Reviewer 2 Report

Comments and Suggestions for Authors

This manuscript introduced the methodology for implementing edge tech with sensor nodes for performance estimation especially forecasting voltage.

1.     It would be evident that edge tech. could forecast solar panel voltage generation. The proposed model predicts solar panel voltage generation based on a single sensor reading.

It is also clear that voltage generation in solar panel is not very sensitive to the environmental conditions so that once the sun light is given even under small amount, the edge reading for voltage is very stable meaning that it is small variation which could be seen easily from the I-V curve. Solar is not a device for voltage but current and power. It is therefore introduction and reading process should be re-examined.

2.     From the format in abstract, all the results should be summarized in the abstract, not the contents, what to do.

3.     Empirical comparisons should be introduced not only voltage but also current and power with all the variables introduced.

4.     All the variables introduced should be identified with proper reasons why this model uses.  Why not wind speed and pressure etc.

5.     Physical explanation should be introduced before the edge sensing technique.

Comments on the Quality of English Language

N/A

Author Response

Dear Reviewer,

Thank you for your review of our manuscript. We have carefully considered your comments and believe that the quality of the paper has improved after incorporating your suggestions.

Below are our responses to your suggestions:

This manuscript introduced the methodology for implementing edge tech with sensor nodes for performance estimation especially forecasting voltage.

2.1.     It would be evident that edge tech. could forecast solar panel voltage generation. The proposed model predicts solar panel voltage generation based on a single sensor reading.

It is also clear that voltage generation in solar panel is not very sensitive to the environmental conditions so that once the sun light is given even under small amount, the edge reading for voltage is very stable meaning that it is small variation which could be seen easily from the I-V curve. Solar is not a device for voltage but current and power. It is therefore introduction and reading process should be re-examined.

We agree with the reviewer, and we are thankful for helping us to improve our paper. In accordance with the comment, we inserted the following part of the text.

“Generally, PV module performance parameters are evaluated based on I-V and P-V curves and numerous other parameters. We used in this research open-circuit voltage (Voc). This is the value of voltage measured under the standard test conditions (STC) or in real-time operating conditions by putting the PV cell terminals in open-circuit conditions and it is measured with a voltmeter when the panel is not connected to any equipment. The value of the voltage in this case is generally higher than the maximum voltage of the panel. There are several reasons for such an approach. First, open-circuit voltage (Voc) is valuable for system planning to avoid overpowering electronics, in this case, it is interesting for the potential design of solar-powered sensor nodes. Second, it is interesting to investigate the influence of light intensity and temperature on solar panel performance and Voc is suitable for this research due to its dependency. The third reason is that we wanted to estimate solar panel behavior based on one output parameter, and we chose the Voc because of all the enlisted reasons.”  

2.2.     From the format in abstract, all the results should be summarized in the abstract, not the contents, what to do.

We agree with the reviewer, and we expanded the abstract with the summarization of the results.

2.3.     Empirical comparisons should be introduced not only voltage but also current and power with all the variables introduced.

We agree with the reviewer, and we are thankful for helping us to improve our paper. Following the comment, we inserted the following part of the text in the Introduction section to explaind this issue as. The explanation as it is answered in comment 2.1.

2.4.     All the variables introduced should be identified with proper reasons why this model uses.  Why not wind speed and pressure etc. 

We agree with the reviewer that the usage of windspeed and other sensors may be useful in this research. It can be covered in the future phases of the research, but the main goal of the research in this phase is to build a methodology for estimating solar panel output based on the most common sensors used as a part of the configuration of sensor nodes. Today, a large number of installed IoT and sensor networks have sensor nodes with sensors presented in this research. Following this reasoning we will allow implementing proposed AI on a broad range of sensor nodes.

2.5.     Physical explanation should be introduced before the edge sensing technique.

We agree with the reviewer, and we inserted the following text in the introduction section.

“Edge intelligence represents the data analysis and solution recommendations at the point where the data is generated or acquired thus saving the time and sometimes bandwidth of the sensor networks. Generally, edge intelligence means the implementation of AI at the far end of IoT and sensor networks, more precisely on sensors nodes and microcontroller boards.”

Comments on the Quality of English Language

N/A

Submission Date

29 December 2023

Date of this review

17 Jan 2024 13:33:11

Reviewer 3 Report

Comments and Suggestions for Authors

Major remarks:

- It is not clearly indicated what the authors' contribution to the field is. It should be specified what exactly is new in their work.

- Using R^2 as the primary quality measure seems inappropriate. This metric is only valid for linear models, such as linear regression, and not for MLP and most other ML techniques.

- What types of error estimation methods are used: cross-validation, bootstrap, train/test, or resubstitution error rate?

- The paper could be improved by using a more diverse dataset, encompassing data from different environmental conditions. The existing dataset seems to have been collected in a brief period and may not adequately represent a broad spectrum of real-world scenarios.

- Although the paper addresses the implementation of machine learning models on edge devices, a more in-depth examination of the balance between model complexity and computational constraints of edge devices, such as the ESP8266, would be beneficial.

- The paper's findings could be bolstered by contrasting the proposed methodology with other current techniques or algorithms for estimating solar panel performance.

- Given that the paper focuses on solar panel performance, an analysis of the system's energy efficiency, particularly the energy usage of edge devices while operating the machine learning models, would be pertinent.

- An elaborate analysis of the system's robustness in handling different types of errors or data anomalies could enhance the paper.

Minor remarks:

- Python libraries should be cited.

- Measures of quality employed for regression analysis should be briefly described in the text. Currently, the paper only presents abbreviations of a few measures.

- Table 1: What does an asterisk next to some metric names signify? How can a negative R^2 be obtained for scenario 8?

- The notation for R^2 varies in the paper, sometimes written as R2. This should be standardized.

- Table 2: The presence of negative values for R^2 and R is unusual.

- Figures 2 and 6 appear very similar. Are both necessary?

- PCA is not a classifier; it is a dimension reduction method.

Author Response

Dear Reviewer,

Thank you for your review of our manuscript. We have carefully considered your comments and believe that the quality of the paper has improved after incorporating your suggestions.

Below are our responses to your suggestions:

Major remarks:

3.1 It is not clearly indicated what the authors' contribution to the field is. It should be specified what exactly is new in their work.

We agree with the reviewer, and we are thankful for helping us to improve our paper. We inserted and altered the following text in the Introduction section to point out our contribution.

“The contribution of this paper presents the methodology for implementing edge intelligence for estimating the potential solar panel performance on the sensor nodes without solar panels. Thus leading to the collection of valuable data for the potential redesign of non-solar-powered sensor nodes, at specific micro-locations, to solar-powered sensor nodes.”

3.2 Using R^2 as the primary quality measure seems inappropriate. This metric is only valid for linear models, such as linear regression, and not for MLP and most other ML techniques.

We agree with the comment that R2 score is not suitable for non-linear regression. We used this score as a part of the model comparison because it is the standard and default method in Scikit-learn library metrics, and a certain number of models in this research belong to the linear regression model family (Linear, Lasso, and ElasticNet, LARS Lasso, Ridge, Bayesian Regression, Automatic Relevance Determination – ARD, Logistic regression, etc.), while some other have significant linear correlation. Also, R2 in Scikit-learn can have negative values in the case of bad models. We used R2 score in combination with other parameters such as MSE and MAE to detect suitable regressors for further implementation of AI. We change the following text to answer this useful comment.

We inserted the following text before Table 1.

“Although generally unsuitable for non-linear regressions, R2 is the standard method in Scikit-learn library metrics, and it can show which used methods can have non-linear results and it can be particularly indicative when R2 has negative values. Also, certain number of models used in further research belong to the linear regression model family. We used R2 score in combination with other parameters such as MSE and MAE to detect suitable regressors for further implementation of AI. The results presented in Table 1 show that the best estimation can be achieved with the Lux sensor (BH1750) in combination with the UV sensor (RMSE 0.08, MAE 0.24, and R2 score 0.9650), and with the Lux sensor only (0.08, 0.22, and 0.97 respectively). The next single sensor with the highest accuracy is the UV sensor (0.1, 0.29, and 0.94 respectively).”

We changed the following text in Section 5. Discussion and 6. Conclusion respectively

“The ESP8266 Random Forest regressor has higher accuracy, with both sensors (MSE around 0.10, MAE 0.06 for UV and 0.03 for Lux, R2 around 0.93). DT has slightly lower accuracy (MSE between 0.13 and 0.14, MAE 0f 0.07 for UV and 0.04 for Lux, and R2 of 0.90 and 0.89 for UV and Lux sensor respectively) in estimating real values compared to RF, but it is still accurate enough for justified implementation of both methods.”

 “The estimation accuracy of RF model is MSE around 0.10, MAE 0.06 for UV and 0.03 for BH1750, and R2 around 0.93 for both UV and BH1750 light sensors. The Decision Tree model has a lower accuracy with MSE between 0.13 and 0.14, MAE 0f 0.07 for UV and 0.04 for BH1750, and R2 of 0.90 and 0.89 for UV and BH1750 sensor respectively. Both metrics justify the usage of the proposed methodology.”

3.3. What types of error estimation methods are used: cross-validation, bootstrap, train/test, or resubstitution error rate?

We have considered this comment as all others. The answer is that we used Scikit-learn metrics with default parameters as it is explained in Scikit-learn documentation. No changes to the default parameters are made.

3.4. The paper could be improved by using a more diverse dataset, encompassing data from different environmental conditions. The existing dataset seems to have been collected in a brief period and may not adequately represent a broad spectrum of real-world scenarios.

We agree with this comment, and it would be helpful for further improvement of our work. So far, we can conclude that there are no third-party solar-related datasets that match our dataset and therefore they cannot be used in this research. According to our knowledge, the data set is unique and tailored for this particular research. 

3.5. Although the paper addresses the implementation of machine learning models on edge devices, a more in-depth examination of the balance between model complexity and computational constraints of edge devices, such as the ESP8266, would be beneficial.

We appreciate this comment, and it would be helpful for the further improvement of our work by analyzing the limitations of implemented AI models on the microcontroller boards. So far, in the current research, we haven't faced any limitations that can distract the full implementation of the methodology. The proposed ESP8266-based microcontroller board has more memory and computational capacities, than Arduino UNO boards. The only limitation is only one analog pin of the ESP8266 microcontroller board, allowing us to connect only one analog sensor at a time. This limitation will be overcome with the implementation of ESP32-based boards. 

3.6. The paper's findings could be bolstered by contrasting the proposed methodology with other current techniques or algorithms for estimating solar panel performance.

We agree with the comment, and we agree the paper’s findings will be enhanced with this addition. The contrast of our approach compared to others in implementing artificial intelligence to estimate values on the sensor nodes that are not adequately configured (e.g. in this particular case to predict solar panel output voltage, without having a solar panel installed). All other solutions make analyses on the PC during or after the data is collected, and the data are collected with the dedicated sensor. According to our knowledge, this solution is unique in the solar-powering focused on the sensor nodes.  By this comment, we altered the text in the Introduction section and abstract to provide more information.

3.7. Given that the paper focuses on solar panel performance, an analysis of the system's energy efficiency, particularly the energy usage of edge devices while operating the machine learning models, would be pertinent.

We appreciate this comment, and it would be helpful for the further improvement of our work by introducing monitoring of energy usage of ESP8266-based and later ESP32-based microcontrollers and clone boards used as sensor nodes. It will be fully implemented in the future, but so far it is only in the experimental phase. The reason is that such platforms capable of monitoring energy usage of the sensor nodes (although it is partly developed), significantly differ in configuration and way of usage, from the platform presented in this paper. Also, an interesting issue in energy consumption monitoring for finding energy-efficient operation modes of ESP8266 and ESP32-based sensor nodes will be examined in the future phases.

3.8. An elaborate analysis of the system's robustness in handling different types of errors or data anomalies could enhance the paper.

This comment is very helpful for the further improvement of our work, but the issue is too complex. The ESP-based boards are mainly made for prototyping but so far they have proved to be a very robust platform. Its robustness is proved as well as our as well other author’s experiences. The analysis of the platform's robustness is a very complex question. In our previous experience, the main issue to be considered is the work of the platform under direct and long-term exposure to sunlight. So, far, the results are good and the platform proves to be prone to heat, but for more accurate results, during this research, we will continue to work on the evaluation of the platform’s stability.

Minor remarks:

3.9. Python libraries should be cited.

We agree with the comment. The references 38-43 for Python libraries are inserted.

3.10. Measures of quality employed for regression analysis should be briefly described in the text. Currently, the paper only presents abbreviations of a few measures.

We expanded the Notification section with an explanation of all regressor types we used in Table 2. Additionally, after Table 2 we inserted a short explanation on measures of quality for regression analyses.

3.11. Table 1: What does an asterisk next to some metric names signify? How can a negative R^2 be obtained for scenario 8?

The asterisk is deleted, it was inserted by mistake. The data acquired with the DHT-22 sensor for humidity in scenario No. 8 does not have a good fitting with the output voltage. So, the R2 score is negative and not suitable for particular data.

3.12 The notation for R^2 varies in the paper, sometimes written as R2. This should be standardized.

We agree with the comment. This is corrected and all scores are changed to R2. It should not be confused with the name of the microcontroller board “Wemos D1 R2”. In the cases of the controller board name, R2 is unchanged. In the case of R2 score, the rest of the text is changed. 

3.13. Table 2: The presence of negative values for R^2 and R is unusual.

We agree with the comment that R2 score is not suitable for non-linear regression. We used this score as a part of the model comparison because it is the standard and default method in Scikit-learn library metrics, and a certain number of models belong to the linear regression model family (Linear, Lasso, and ElasticNet, LARS Lasso, Ridge, Bayesian Regression, Automatic Relevance Determination – ARD, Logistic regression, etc.), while some other have significant linear correlation. Also, R2 in Scikit-learn can have negative values in the case of bad models. We used R2 score in combination with other parameters such as MSE and MAE to detect suitable regressors for further implementation AI. We changed the following parts of the text to answer this useful comment as it is written in 3.2. 

3.14. Figures 2 and 6 appear very similar. Are both necessary?

Yes, those figures look very similar but are not the same. Fig. 2 represents a data acquisition full-feature platform for acquiring data for building estimation models. Fig. 6 shows a sensor network platform with implemented AI, and reduced component set, without solar panels and optionally without ambient temperature and humidity sensors. Fig. 6 is changed as explained and differs from Fig. 2. The figure caption and text before the figure are also changed to be more descriptive. 

3.15. PCA is not a classifier; it is a dimension reduction method.

We agree with the comment and we are thankful for helping in correcting our mistakes. We deleted PCA from text.

Submission Date

29 December 2023

Date of this review

12 Jan 2024 14:21:18

Round 2

Reviewer 2 Report

Comments and Suggestions for Authors

Most of the comments have been applied to revised manuscript. Hope  this research would be helpful in expecting and stabilizing the PV power based grid in the country.

Comments on the Quality of English Language

N/A

Reviewer 3 Report

Comments and Suggestions for Authors

The authors responded meaningfully to all comments and made the necessary corrections to the article. In my opinion, the work is suitable for publication.